# Sexual Dimorphisms and Asymmetries of the Thalamo-Cortical Pathways and Subcortical Grey Matter of Term Born Healthy Neonates: An Investigation with Diffusion Tensor MRI

**DOI:** 10.3390/diagnostics11030560

**Published:** 2021-03-20

**Authors:** Fadoua Saadani-Makki, Ardalan Aarabi, Mahshid Fouladivanda, Karman Kazemi, Malek Makki

**Affiliations:** 1PeriTox, UMR-I 01, INERIS, University of Picardy Jules Verne, 80037 Amiens, France; fsaadani.makki@gmail.com; 2Laboratory of Functional Neuroscience and Pathologies (LNFP), School of Medicine, University of Picardy Jules Verne, 80037 Amiens, France; ardalan.aarabi@u-picardie.fr; 3Department of Electrical and Electronics Engineering, Shiraz University of Technology, Shiraz 71557-13876, Iran; m.fouladivanda@sutech.ac.ir (M.F.); kazemi@sutech.ac.ir (K.K.)

**Keywords:** neonates, brain, DTI, putamen, caudate nucleus, globus pallidus, thalamus, thalamo-cortical pathway, asymmetry, boys, girls

## Abstract

Diffusion-tensor-MRI was performed on 28 term born neonates. For each hemisphere, we quantified separately the axial and the radial diffusion (AD, RD), the apparent diffusion coefficient (ADC) and the fractional anisotropy (FA) of the thalamo-cortical pathway (THC) and four structures: thalamus (TH), putamen (PT), caudate nucleus (CN) and globus-pallidus (GP). There was no significant difference between boys and girls in either the left or in the right hemispheric THC, TH, GP, CN and PT. In the combined group (boys + girls) significant left greater than right symmetry was observed in the THC (AD, RD and ADC), and TH (AD, ADC). Within the same group, we reported left greater than right asymmetry in the PT (FA), CN (RD and ADC). Different findings were recorded when we split the group of neonates by gender. Girls exhibited right > left AD, RD and ADC in the THC and left > right FA in the PT. In the group of boys, we observed right > left RD and ADC. We also reported left > right FA in the PT and left > right RD in the CN. These results provide insights into normal asymmetric development of sensory-motor networks within boys and girls.

## 1. Introduction

The thalamus and the basal ganglia (caudate, putamen and globus pallidus) are deep gray matter structures known to modulate motor, cognitive, and sensory functions [1,2]. However, the micro-structural changes of the subcortical regions differ across the thalamus (TH), the caudate nucleus (CN), the putamen (PT) and the globus pallidus (GP) [3]. The maturation patterns of membrane proliferation [4] and fiber myelination [5] run in parallel to the brain circuit formation. These phenomena are intense in the developing deep grey matter of the neonate brain from 36 to 43 post menstrual weeks and result from regulated molecular and cellular processes [6]. Synapse formation [7], dendritic arborization [8], neurogenesis, neuronal migration, axonal growth and pruning [9] increase the brain size exponentially to reach 90% of adult volume around 2 years of age [10]. It has been reported that the whole-brain growth is about 1% per day, slowing to 0.4%/day by the end of the first 3 months [11]. The formation and the development of neural connections between the cerebral cortex and the thalamus require thalamo-cortical (THC) projections to synapse transiently in the temporary cortical subplate before penetrating the cortical plate [12]. Functional and structural MRI connectivity analysis between the TH and the cortex has described a parcellation of the TH with precise connectivity to specific cortical regions [13]. The thalamo-cortical fibers and the cortical dendritic tree are related to the development of synchronous cortical development in the last trimester when the subplate is at maximal extent [14,15]. At 24 weeks post-conception age, brain development is characterized by the migration of thalamo-cortical afferents from the subplate into the overlying cerebral cortex [16]. During this critical phase, any damage to these cell populations could impact the microstructural development of the subcortical gray matter and the cortex, thus disrupting the connective white matter fibers [17,18]. These connections form a set of parallel and segregated cortico-thalamic loops that project to the majority of the cortex and act as a link between the basal ganglia and cortex [19,20]. This allows the information flow to drive ascending integration of sensory information into higher-order cortical networks [21,22,23]. The thalamo-cortical network features widespread connectivity throughout the TH involving the primary sensory motor, temporal, medial prefrontal, anterior cingulate, and fronto-parietal insular cortices, and others are mainly limited to their areas of dominance (primary auditory, primary visual and lateral parietal) [12].

Any anatomical alterations underlying the subplate neurons, thalamic neurons and pre-myelinating oligodendrocytes that result from hypoxia/ischemia and infection/inflammation ultimately affect the corresponding neurological functions [24,25]. Several studies demonstrated that impaired cognition in premature infants is associated with impaired connectivity in the frontal-subcortical pathways including the THC networks leading to neurological disorders (attentional deficit/hyperactivity disorder (ADHD), autism spectrum disorders (ASD) and learning disabilities [21,26,27,28,29,30]. Many neonate studies provided evidence that during the second and third trimester, the brain undergoes substantial reorganization with exponential increases of the cortical volume and surface [31,32]. Diffusion MRI studies have shown that the microscopic tissue coherence decreases with the increases of anisotropy [22,33]. A histology study by Mrzljak et al. (1992) [34] demonstrated that this loss in cortical coherence corresponds to neuronal differentiation, dendritic arborization, and cortical axonal ingrowth. Previous studies on premature infants have reported asymmetric volumes in the perirhinal cortex [35], Brodmann area A44 [35,36], primary motor cortex [37], the middle temporal visual [38] and the language areas [39,40]. Studying the neonate brain is particularly challenging because the physiologic patterns change dramatically, the degree of plasticity is high, the structure sizes are too small, the WM/GM contrast is low, and there is motion of artifacts if sedation is not administrated. The plasticity and capacity for adaptation of the human brain offers considerable potential for optimizing brain outcomes through the development of early diagnostic tools and early interventions [41,42,43]. Noninvasive neuroimaging such as MRI is of utmost to study and monitor prenatal and postnatal brain development to improve diagnostic accuracy and to plan better guiding circuit-specific early interventions.

The aim of this study was to extend the previous work in lateralization of the cortico-spinal tracts [44] to provide insight into subcortical grey matter and THC pathways. Using DTI, we investigated whether 1) there are microstructural asymmetries in the CN, PT, GP, TH and THC pathways can be observed in neonates under 1 month of age and 2) whether there is a gender effect in these subcortical areas. For each structure, we assessed the sexual dimorphisms and left–right asymmetries for each DTI variable. Providing normative data to evaluate the white matter integrity in term born neonates will help in identifying abnormal brain development and pathological conditions.

## 2. Materials and Methods

### 2.1. Subjects

Thirty-five healthy term born neonates were recruited from the postnatal ward at our hospital and were examined with MRI. The subjects presented here were part of a prospective clinical trial investigating neonates with congenital heart disease undergoing transposition of the great artery. The institutional ethics committee approved this study, and written informed consent was obtained from guardians. The inclusion criteria are gestation age >37 weeks and normal Apgar score. Exclusion criteria are any congenital malformation or small for 37 weeks of gestation. Selected neonates were scanned in natural sleep and were monitored by pulse oximetry. Seven neonates were excluded either because of motion artefacts or they did not fall asleep to achieve the examination. Overall, DTI was successfully achieved on 28 infants (13 girls). A neonatal nurse was present during the examination to check the oxygen saturation, the arterial blood pressure, the heart rate and P_CO_2__. Demographic details and age at scanning are listed in Table 1.

A routine clinical neuroimaging examination was performed on a 3-T scanner (GE Healthcare) with an eight-channel head coil. It included (i) isotropic 3D T1-weighted images utilizing a magnetization preparation fast spoiled gradient echo, (ii) sagittal 2D T2-weighted Fast-Spin-Echo, (iii) high-resolution morphologic axial T_2_W fast spin echo, and (iv) diffusion-tensor spin-echo (DTI) echo planar sequence. The DTI was performed using 35 non-collinear gradient-diffusion directions and one volume of reference T2W images. The diffusion sensitivity gradient was set to b value 700 sec/mm^2^. Imaging parameters are slice thickness 2.5 mm (between 34 and 38 slices covering the whole brain), FOV 220 mm^2^, matrix 128 × 128 homodyne reconstructed in 256 × 256 to achieve an in-plane resolution of 0.85 × 0.85 mm^2^. Pulsed gradient double refocusing pulses were used to reduce eddy-current artifacts and parallel imaging (ASSET × 2) was applied. The acquisition time was ∼5 min 30 s.

### 2.2. Image Acquisition and Analysis

#### 2.2.1. Image Processing

The thalamo-cortical fiber (THC) tracking was carried out with DTI *Studio* software (Department of Radiology, Johns Hopkins University, Baltimore, MD, USA) using the deterministic FACT algorithm (fiber assignment continuous tracking) and the following two rejection criteria: fractional anisotropy (FA) < 0.15 or angulation > 70°. The seed regions were manually drawn on each axial slice of the TH on FA maps (4 to 5 slices). It has been applied with the inclusive “OR” operator. The target regions were selected FA maps at the coronal plane at the level of the genu. This was performed with an exclusive operator “AND”. It was reproduced bilaterally to determine left and right THC pathways. The axial diffusion (AD), the radial diffusion (RD), the apparent diffusion coefficient (ADC) and the FA were measured and averaged separately over the entire left and the entire right THC. For each group (girls, boys and combined) we measured the DTI variables bilaterally on the selected four structures: TH, PT, GB, and CN (Figure 1).

For structure analysis, the delineation of each region was performed manually on each hemisphere. The regions of interest were carefully drawn covering the entire surface area on FA maps because the structure edges are well displayed and easily identified. An experienced scientist with over 10 years of practice carried out the ROI drawings. This procedure was repeated twice (one month apart) to minimize the subjectivity error. The results of the two measures were averaged to perform the statistical comparisons. The delineation was reproduced on axial planes and going through the entire structure. We calculated the DTI variables (AD, RD, ADC and FA) by averaging the measured values. For each of the 4 DTI variables, we calculated the asymmetry index (AI) to identify the dominance hemisphere per structure and gender. Dominance was attributed to the left side if AI value is positive and to the right if it is negative.
(1)$$AI=\frac{2\cdot \left(L-R\right)}{\left(L+R\right)}\%$$

#### 2.2.2. Statistical Data Analysis

The statistical analyses were performed with SPSS (Statistical Package for the Social Sciences, Version 22.0). The inter-group differences (Girls vs. Boys) were tested separately for the 5 structures (TH, PT, GP, CN and THC) on both sides (left and right). This test was repeated for each DTI index using a general linear model multiple analysis of covariance with *p* < 0.05 were considered significant. Postconceptional age at MR imaging of the subjects was included as a covariate to control for age-related changes of DTI indices. Following Bonferroni correction for type I error (5 structures × 2 sides × 2 groups = 20 tests), group differences with *p* < 0.002 were considered significant.

The intra-group statistical comparison between left and right was performed separately for each structure (TH, PT, GP, CN and THC) and each group (girls, boys and combined) with paired *t*-Test. The significance was set to *p* < 0.05.

## 3. Results

### 3.1. Left-Right Asymmetry Girls and Boys Combined

Left and right comparisons of the combined group (girls and boys) demonstrated significantly higher axial diffusion (*p* = 0.009) and ADC (*p* = 0.023) in the right thalamus compared to the left (Figure 2A). The PT has significantly higher FA in the left (*p* = 0.015) compared to the right. No asymmetry was reported in the globus pallidus (Table 2). The right THC pathway exhibited significantly larger axial diffusion (*p* = 0.003), radial diffusion (*p* = 0003) and ADC (*p* = 0.001) compared to the left (Figure 3A). The left caudate nucleus had higher radial diffusion (*p* = 0.049) and higher ADC (*p* = 0.040) compared to the right (Figure 4A).

### 3.2. Left–Right Asymmetry in the Girl Group

The thalamus of the girl group had significantly higher axial diffusion in the right (*p* = 0.048) compared to the left (Figure 2B) and the right THC pathway presented with significantly higher axial diffusion (*p* = 0.004), radial diffusion (*p* = 0.010) and ADC (*p* = 0.001) (Figure 3B). The FA of the left PT was significantly higher (*p* = 0.015) than the right. The CN and the GP did not reveal any significant asymmetry (Table 3).

### 3.3. Left–Right Asymmetry in the Boy Group

Significantly higher radial diffusion (*p* = 0.047) and ADC (*p* = 0.005) were recorded in the right compared to the left TH (Figure 2C) in the boys group (Table 4). The FA of the left PT was significantly higher (*p* = 0.044) than the right (Figure 5). The caudate nucleus showed significantly higher ADC in the right (*p* = 0.043) compared to the left (Figure 4B). The GP did not reveal any significant asymmetry.

### 3.4. Hemispheric Dominance by Group

Overall, the combined group and the boy group had right dominance of the thalamic RD, but the girl group showed a left dominance. The dominances of the AD, ADC and FA were all to the right. In the THC, the right dominance was the same for all groups and each DTI variable. In the PT, the AD dominance was to the left in the combined group and among girls, but it was to the right side among boys. We observed the opposite trend in ADC, where the dominance was to the right for girls and the combined group and to the right among the boys. In addition, the dominance of ADC was to the right while the FA dominance was to the left (Table 5).

In the CN we observed left dominance in AD, RD and ADC among girls and boys and in the combined group. The FA of the CN was left dominant among girls and right dominant among boys and the combined group.

The GB dominance side was in the left for RD and ADC in all three groups. For AD the dominance was to the right for girls and to the left for boys and the combined group. The FA was dominant in the right for girls, to the left for boys and to the right for the combined group.

### 3.5. Gender Dimorphism

The GLM analysis of covariance of the DTI variables revealed that there was no significant difference between boys and girls in any of the structures (Table 6). No gender dimorphism was observed in the left or in the right side.

## 4. Discussion

This study on the neonate brain aimed to answer two questions regarding the thalamus, basal ganglia and the thalamo-cortical connectivity. The first involved the gender effect on their cyto-architecture. In fact, we did not observe any sexual dimorphism with regard to DTI variables neither in the left nor in the right side of these structures and the THC pathways. The second question was about left–right asymmetry in these parts of deep grey matter. Our results showed different findings by side, gender, structure and index. There was right ward asymmetry in all DTI indices (right > left) in the thalamo-cortical pathways of boys, girls and the combined group. We measured also right ward asymmetry in all DTI metrics in the thalamus of the combined group, boys, and girls but the RD of the latest. In the putamen, there was left asymmetry in the AD and FA, and right asymmetry in the RD of the three groups. The ADC exhibited different asymmetry: Left among boys, right among girls and the combined group. The caudate nucleus asymmetry was exclusively to the left among girls. The combined group and the boy group had left asymmetry in AD, RD and ADC but right asymmetry FA. In the globus-pallidus, the boy group had left asymmetry in all DTI indices while girls had right asymmetry in AD and FA, but left asymmetry in RD and AD. The combined group exhibited left asymmetry in AD, RD and ADC but right asymmetry in FA.

In full-term neonates GA, birth weight, crown-heel length and head circumference affect WM maturation and it has been demonstrated that there is a biological link between the DTI indices (FA, ADC, AD and RD) and brain WM microstructural integrity, water diffusivities, axonal growth and myelination [45,46]. The thalamus and the basal-ganglia experience significant and continuous changes with age and they all reach the 90% mark between the ages of 21 and 24 years [47]. In our study, the patterns of cerebral asymmetry of the selected brain structures are different from those observed in other structures, suggesting that neonate patterns of cerebral asymmetry after birth are not homogeneous and might change until they reach the milestone mark. Cerebral asymmetry has been associated with lateralization including handedness, dexterity, and language abilities [48]. Several neurological disorders are believed to be developmental and are associated with atypical brain asymmetry [49], particularly schizophrenia [50], autism [51], Tourette syndrome [52] and Sturge–Weber syndrome [53]. The anatomical and functional asymmetries in the brain are thought to arise from asymmetric gene expression in the embryonic human cortex [54,55], and can be observed in the human embryonic cortex around 12 weeks [54]. Previous investigations observed left greater than right lateral ventricles in the fetus brain [56] hemisphere about 20–22 weeks gestational age [57], and neonatal volumetric cortical white matter [58]. Other studies on children and adults reported right greater than left hemisphere [10,39,59]. Neonates with early deviant brain asymmetry who are at risk for a neurodevelopmental disorder may influence the pattern of asymmetry reported on children and adolescents because it is likely an early environmental insult to the developing brain. Puetz VB et al. (2017) [60] suggested that postnatal stress disorder with loss of normal frontal lobe asymmetry may influence the development of adult patterns of asymmetry.

Previous neuroimaging investigation of the potential neuroanatomical substrates of cognitive impairment from various causes supported the involvement of the thalamus in the network responsible for cognitive performance [61]. The thalamus serves as a relay station for all cortical sensory systems and plays a critical and a central role in the development of cortical circuitry and multisensory and sensorimotor functions and an isolated damage of the thalamus may result in cognitive dysfunction. Bültmann et al. (2017) [62] demonstrated that the TH structure has high concentration of myelin-associated macromolecules and a low water content, low diffusivities, and low coherence. The interconnections between the TH and PT have been demonstrated to belong to the motor circuit [63,64], and the oculomotor circuits including the dorsolateral prefrontal subcortical circuit (DLPC), the lateral orbitofrontal-subcortical circuit (LOFC). The vulnerability between the TH and PT related to preterm birth has been demonstrated by the scale and the translation of the left PT with that of the left TH while the right TH and PT are significantly associated through rotation [65,66]. The TH significantly right greater than left AD and ADC in the combined group, RD and ADC among boys, and AD among girls are in line with these interactions.

With regards to the caudate, it has been shown that the left is larger than the right, while the right hippocampus, thalamus, putamen and cerebellum were larger than the left [11,35,67]. Koolschijn and Crone (2013) have found statistically significant shifts toward increased volumetric and surface area measurements among males relative to females in line with previous findings [68]. In our study, left larger than right RD were observed in the combined group and among boys. The left ADC was larger than the right in the combined group as well. Hypoxic-ischaemic injury studies have documented atrophy of basal ganglia structures [69] and thalamus [70]. Voxel-based morphometry confirmed that hypoxic ischaemic patients showed bilateral volume reduction in the thalamus and the caudate nucleus [71]. Alcauter et al. (2017) study on neonates identified the development of the thalamo–sensorimotor and the thalamo–salience network connectivity as the only higher-order network that can be tracked before 1 year of age and an alteration of the thalamic-cortical connectivity was associated to cognitive impairment in premature newborns [72].

During the neonate period, changes in DTI metrics correlate with any of the following phenomena: increase in myelin sheaths, increments of axon density, cell membrane, and glial cells [46]. The globus-pallidus reflects development of postural balance. However, Cahill-Rowley et al. (2019) did not report a significant correlation between the DTI metrics of the globus pallidus and both walking velocity and single limb support [73]. The lack of significant asymmetry observed in the globus-pallidus may presumably relate to slow development of sensorimotor ability. This feature might change with age and the neurodevelopmental process but given the complexity in brain development and the environmental relationship, further investigation is needed to explain the physiological and neurological processes as deep gray matter structures show very large percent changes and the white matter structures showing smaller, but still significant, changes [45].

Around 26 GW the fetal brain development is characterized by the establishment of structural THC axons from the cortical plate to subplate [9,74] and the beginning of massive cortical synaptogenesis, which marks the first step towards the beginning of sensory driven activity of neuronal circuits. Our results of the thalamo-cortical pathway demonstrated significantly right larger than left AD, RD and ADC in the girl group and in the combined group. The asymmetry in the boy group did not reach the significant level. The reported structural asymmetries are in line with previous findings showing that the correlation of WM with GA is more pronounced in the left hemisphere [75] and the lateralization of the somatosensory can be observed at birth [76]. The lateralization might be attributed to genetic programs during the prenatal stage, as asymmetry of gene expression in the human embryonic cortex has been found at gestational of 12 weeks [76].

The pattern of sexual dimorphisms in this period of brain development are generally different from those observed in later stages of brain development, suggesting that adult patterns of sexual dimorphism arise after birth and persist in adult brain development. Interestingly, there was no gender regional specificity in the selected gray matter structures, thalamus and thalamo-cortical pathways that can be identified with DTI. Sexual dimorphism has been observed in the neonatal brain after birth with males having ~9% larger intracranial volume than females, about 10% more cortical gray matter than females and 6% more cortical white matter [58]. In addition, previous fetal neonate studies have reported sexual dimorphism in head circumference as early as the second trimester [77]. The two sides and four-indices data analysis failed to demonstrate any sexual dimorphism in the selected structures.

## 5. Conclusions

This investigation showed that there is no significant boy–girl difference in the cyto-architecture of the basal ganglia, thalamus and thalamo-cortical pathways. It is noteworthy to add that we measured different microstructural growth evidence of these structures among these structures in boys and girls. Subtle left-right asymmetric differences of these deep grey-matters have been observed within the first 3–4 weeks after birth. These are the results of genetic programming, but the neurodevelopment functioning might be affected by the social environment across the lifespan. Although this study has a limited number of subjects, we suggest that structural impairment of the thalamic asymmetry might be an objective marker of cognitive and motor functioning in children. Further investigations involving both larger sample size and multimodal imaging are required to confirm this suggestion. Our findings in mature neonate girls and boys set the ground truth data for future studies that associate neuro-behavior and subcortical sensory, motor and cognitive networks. Our study provides a useful baseline for future research toward detecting and characterizing a variety of pathological conditions as departures from expected growth trajectories.

## Figures and Tables

**Figure 1 diagnostics-11-00560-f001:**
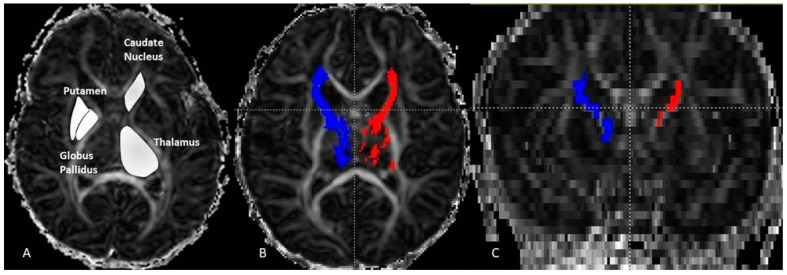
Axial fractional anisotropy (FA) image (**A**) displaying the ROI manual drawing on the thalamus, caudate-nucleus, putamen and globus pallidus. On the (**B**) image the result of fiber tracking of the left (red) and right (blue) thalamo-cortical pathways. The same pathways are displayed in the coronal plane (**C**). These were performed on a neonate girl (GA = 43 weeks + 3 days).

**Figure 2 diagnostics-11-00560-f002:**
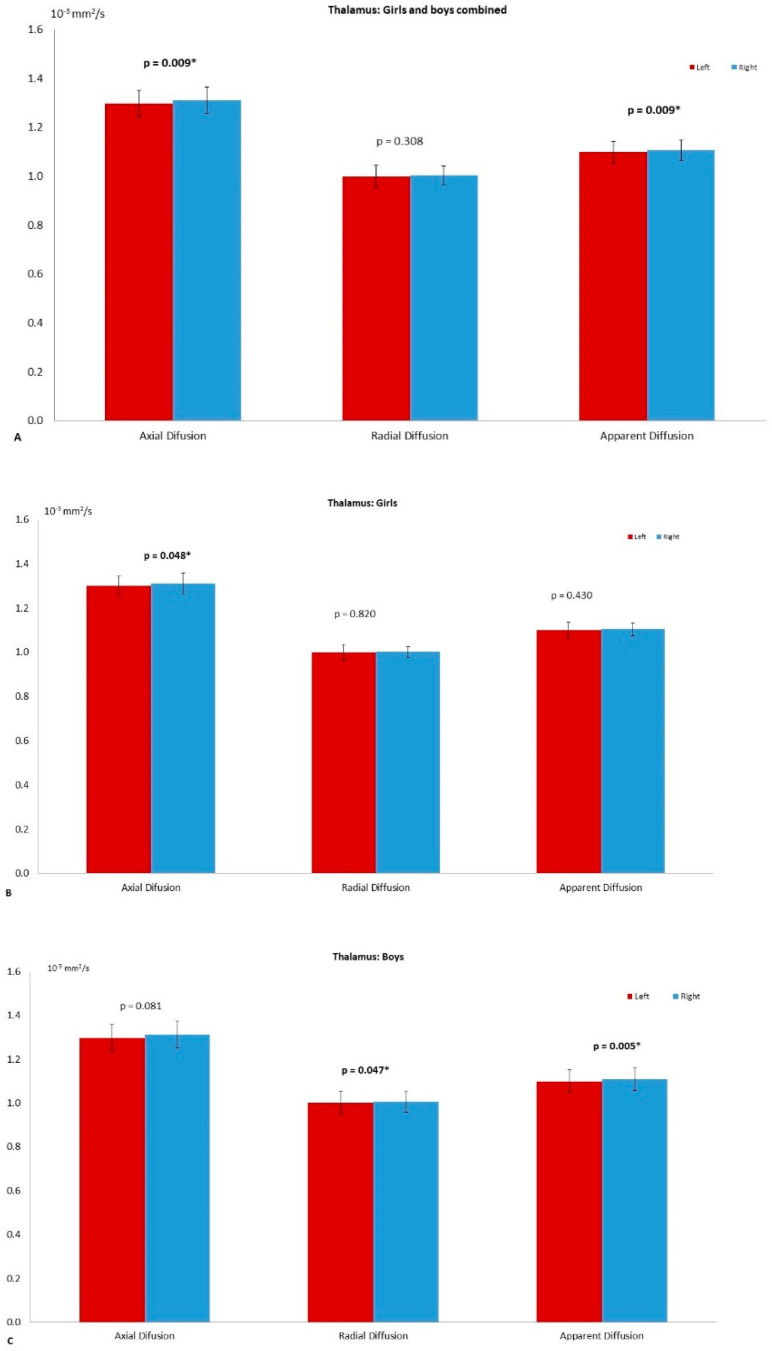
The left–right asymmetry of the thalamus. (**A**) The differences on axial diffusion (AD, *p* = 0.009), radial diffusion (RD) and apparent diffusion coefficient (ADC, *p* = 0.023) in the combined group (girls and boys). The star (*) refers to significant difference. On (**B**) the asymmetry observed in the girl group with AD (*p* = 0.048), RD and ADC. The graph on (**C**) shows the differences for the boy group AD, RD (*p* = 0.047) and ADC (*p* = 0.005). The significance was set to *p* = 0.05.

**Figure 3 diagnostics-11-00560-f003:**
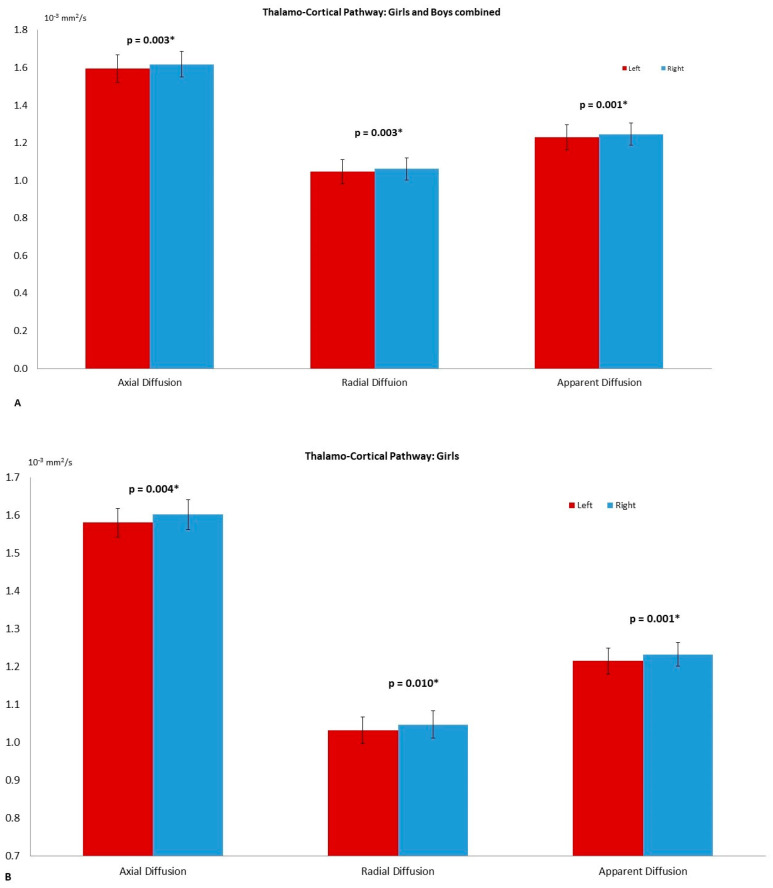
The asymmetry observed in the thalamo-cortical pathway with the axial diffusion (AD, *p* = 0.003); radial diffusion (RD, *p* = 0.003) and apparent diffusion coefficient (ADC, *p* = 0.001) of the combined group (**A**). On (**B**) the differences among girls AD (*p* = 0.004), RD (*p* = 0.010) and ADC (*p* = 0.001). The significance was set to *p* = 0.05 and highlighted with a star superscript (*).

**Figure 4 diagnostics-11-00560-f004:**
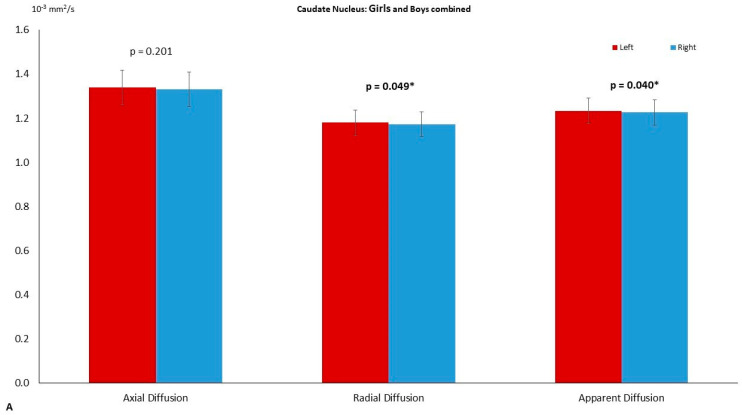
The left–right asymmetry of the caudate-nucleus. (**A**) The differences on axial diffusion (AD), radial diffusion (RD, *p* = 0.049) and apparent diffusion coefficient (ADC, *p* = 0.040) in the combined group (girls and boys). On (**B**) the asymmetry observed in the boy group with RD (*p* = 0.043). The significance was set to *p* = 0.05 and highlighted with a star superscript (*).

**Figure 5 diagnostics-11-00560-f005:**
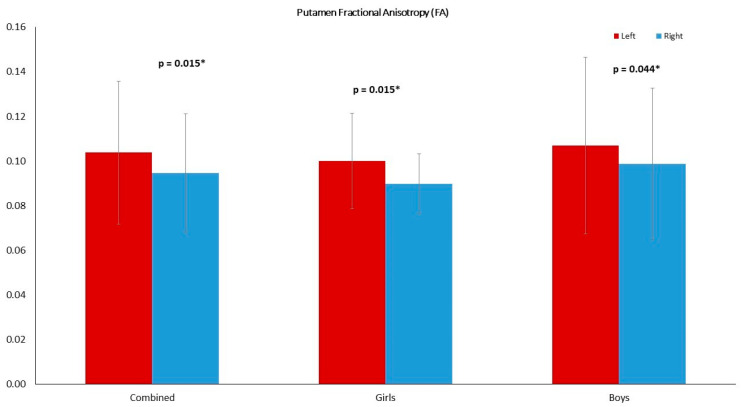
The putamen FA was higher in the left compared to the right in all groups: girls and boys combined (*p* = 0.015), girls (*p* = 0.015) and boys (*p* = 0.044). The significance was set to *p* = 0.05. and highlighted with a star superscript (*).

**Table 1 diagnostics-11-00560-t001:** Descriptive gestation age (GA) statistics of the neonates per group (W refers to number of week and D number of days).

Gestation Age (GA)	N	Minimum	Maximum	Mean	Median
Girls + Boys combined	28	38 W + 3 D	44 W + 1 D	41 W ± 6 D	42 W + 0 D
Girls	13	38 W + 3 D	43 W + 3 D	41 W ± 6 D	42 W + 6 D
Boys	15	38 W + 3 D	44 W + 1 D	41 W ± 6 D	42 W + 0 D

**Table 2 diagnostics-11-00560-t002:** Paired *t*-Test of the combined group (Girls + Boys). This was performed to compare left and right values of the axial diffusion (AD), radial diffusion (RD), apparent diffusion coefficient (ADC) and fractional anisotropy (FA). The comparison was carried out on the thalamo-cortical pathways, the thalamus, caudate-nucleus, globus pallidus and putamen. * The difference was set to be significant when it was below 0.05.

Girls + Boys	AD 10^−3^ mm^2^/s		RD 10^−3^ mm^2^/s		ADC 10^−3^ mm^2^/s		FA	
	Left	Right	Left	Right	Left	Right	Left	Right
Thalamo-Cortical								
Mean	1.596	1.617	1.047	1.0681	1.230	1.247	0.264	0.265
SD	±0.073	±0.068	±0.065	±0.059	±0.066	±0.060	±0.015	±0.017
L vs. R (p)	0.003		0.003		0.001		0.772	
Thalamus								
Mean	1.298	1.311	0.999	1.003	1.099	1.106	0.177	0.182
SD	±0.054	±0.055	±0.045	±0.038	±0.045	±0.042	±0.017	±0.015
L vs. R (p)	0.009		0.308		0.023 *		0.116	
Putamen								
Mean	1.311	1.303	1.119	1.128	1.183	1.186	0.104	0.095
SD	±0.078	±0.066	±0.054	±0.049	±0.054	±0.048	±0.032	±0.026
L vs. R (p)	0.597		0.086		0.267		0.015 *	
Caudate Nucleus								
Mean	1.340	1.332	1.180	1.173	1.233	1.226	0.085	0.085
SD	±0.078	±0.078	±0.057	±0.057	±0.058	±0.057	±0.029	±0.027
L vs. R (p)	0.201		0.049		0.040		0.939	
Globus Pallidus								
Mean	1.376	1.372	1.080	1.071	1.179	1.171	0.163	0.164
SD	±0.070	±0.064	±0.050	±0.054	±0.051	±0.052	±0.024	±0.025
L vs. R (p)	0.906		0.06		0.191		0.298	

**Table 3 diagnostics-11-00560-t003:** Paired *t*-Test of the two separate groups by gender Girls. The test was performed to compare left and right values of the axial diffusion (AD), radial diffusion (RD), apparent diffusion coefficient (ADC) and fractional anisotropy (FA). The comparison was carried out on the thalamo-cortical pathways, the thalamus, caudate-nucleus, globus pallidus and putamen. The difference was set to be significant when it was below 0.05.

Paired *t*-Test		AD 10^−3^ mm^2^/s			RD 10^−3^ mm^2^/s			ADC 10^−3^ mm^2^/s			FA		
		Left	Right	*p*	Left	Right	*p*	Left	Right	*p*	Left	Right	*p*
Girls													
TH-C	M	1.579	1.601	0.004	1.032	1.047	0.010	1.215	1.232	0.001	0.266	0.267	0.821
	SD	0.037	0.039		0.035	0.036		0.034	0.031		0.136	0.019	
TH	M	1.300	1.311	0.04	0.999	1.001	0.820	1.099	1.104	0.430	0.179	0.185	0.068
	SD	0.096	0.046		0.035	0.036		0.036	0.029		0.012	0.015	
PT	M	1.296	1.292	0.597	1.114	1.128	0.086	1.175	1.182	0.267	0.100	0.089	0.015
	SD	0.041	0.028		0.039	0.035		0.035	0.031		0.021	0.013	
CN	M	1.338	1.322	0.090	1.171	1.166	0.539	1.226	1.218	0.223	0.089	0.084	0.134
	SD	0.047	0.036		0.034	0.039		0.032	0.033		0.023	0.020	
GP	M	1.358	1.359	0.906	1.068	1.056	0.067	1.165	1.157	0.191	0.162	0.167	0.298
	SD	0.036	0.035		0.031	0.036		0.029	0.032		0.017	0.019	

**Table 4 diagnostics-11-00560-t004:** Paired *t*-Test of the two separate groups by gender Girls and Boys. The test was performed to compare left and right values of the axial diffusion (AD), radial diffusion (RD), apparent diffusion coefficient (ADC) and fractional anisotropy (FA). The comparison was carried out on the thalamo-cortical pathways, the thalamus, caudate-nucleus, globus pallidus and putamen. * The difference was set to be significant when it was below 0.05.

Paired *t*-Test		AD 10^−3^ mm^2^/s			RD 10^−3^ mm^2^/s			ADC 10^−3^ mm^2^/s			FA		
		Left	Right	*p*	Left	Right	*p*	Left	Right	*p*	Left	Right	*p*
Boys													
THC	M	1.061	1.632	0.085	1.060	1.074	0.083	1.244	1.260	0.070	0.262	0.263	0.862
	SD	0.095	0.086		0.083	0.074		0.086	0.077		0.017	0.016	
TH	M	1.297	1.312	0.081	1.002	1.006	0.047	1.100	1.108	0.005	0.175	0.180	0.3345
	SD	0.062	0.032		0.052	0.047		0.052	0.050		0.020	0.015	
PT	M	1.324	1.313	0.338	1.124	1.128	0.662	1.191	1.189	0.864	0.107	0.099	0.044
	SD	0.100	0.087		0.066	0.059		0.066	0.059		0.039	0.034	
CN	M	1.342	1.340	0.801	1.188	1.177	0.043 *	1.240	1.232	0.079	0.080	0.086	0.279
	SD	0.096	0.099		0.070	0.067		0.072	0.071		0.032	0.032	
GP	M	1.392	1.393	0.392	1.090	1.084	0.351	1.190	1.184	0.287	0.165	0.162	0.443
	SD	0.087	0.080		0.061	0.064		0.063	0.063		0.028	0.029	

**Table 5 diagnostics-11-00560-t005:** Asymmetry index (AI) between left and right of the selected structures and pathway. It was calculated for each index (AD axial diffusion, RD radial diffusion, ADC apparent diffusion coefficient and FA fractional anisotropy) and group (girls, boys, combined). The letter in parenthesis indicates the dominant side (left L, right R).

AI %		AD	RD	ADC	FA
Thalamo-cortical	Girls	−1.37 ± 1.42 (**R**)	−1.45 ± 1.79 (**R**)	−1.43 ± 1.26 (**R**)	−0.20 ± 4.03 (**R**)
	Boys	−1.23 ± 2.73 (**R**)	−1.23 ± 2.88 (**R**)	−1.22 ± 2.63 (**R**)	−0.38 ± 4.53 (**R**)
	Combined	−1.37 ± 2.12 (**R**)	−1.41 ± 2.27 (**R**)	−1.40 ± 1.91 (**R**)	−0.18 ± 3.88 (**R**)
Thalamus	Girls	−1.12 ± 2.02 (**R**)	0.06 ± 2.44 (**L**)	−0.40 ± 1.73 (**R**)	−4.40 ± 9.26 (**R**)
	Boys	−0.85 ± 1.76 (**R**)	−0.93 ± 1.63 (**R**)	−0.89 ± 1.05 (**R**)	−1.37 ± 8.66 (**R**)
	Combined	−0.97 ± 1.89 (**R**)	−0.48 ± 2.11 (**R**)	−0.67 ± 1.43 (**R**)	−2.73 ± 9.07 (**R**)
Putamen	Girls	0.06 ± 2.04 (**L**)	−1.50 ± 2.18 (**R**)	−0.92 ± 1.84 (**R**)	10.14 ± 12.78 (**L**)
	Boys	0.76 ± 3.37 (**L**)	−0.35 ± 2.44 (**R**)	0.076 ± 2.46 (**L**)	7.36 ± 13.56 (**L**)
	Combined	0.54 ± 2.82 (**L**)	−0.75 ± 2.38 (**R**)	−0.27 ± 2.25 (**R**)	8.52 ± 12.80 (**L**)
Caudate Nucleus	Girls	1.01 ± 2.40 (**L**)	0.30 ± 0.02 (**L**)	0.56 ± 1.69 (**L**)	4.87 ± 16.08 (**L**)
	Boys	0.16 ± 2.09 (**L**)	0.90 ± 1.56 (**L**)	0.63 ± 1.27 (**L**)	−5.84 ± 20.67 (**R**)
	Combined	0.56 ± 2.24 (**L**)	0.61 ± 1.58 (**L**)	0.59 ± 1.45 (**L**)	−0.87 ± 19.13 (**R**)
Globus Pallidus	Girls	−0.38 ± 2.30 (**R**)	0.99 ± 2.09(**L**)	0.45 ± 1.65 (**L**)	−3.79 ± 10.87 (**R**)
	Boys	0.82 ± 2.87 (**L**)	0.55 ± 2.13 (**L**)	0.66 ± 2.00 (**L**)	1.62 ± 9.22 (**L**)
	Combined	0.27 ± 2.57(**L**)	0.84 ± 2.08 (**L**)	0.62 ± 1.84 (**L**)	−0.59 ± 9.66 (**R**)

**Table 6 diagnostics-11-00560-t006:** Inter-group differences Girls vs. Boys were carried out separately for the five structures using a general linear model analysis of covariance with gestation age at MRI as the covariate. The test was performed bilaterally to compare the values of the axial diffusion (AD), radial diffusion (RD), apparent diffusion coefficient (ADC) and fractional anisotropy (FA). The difference was set to be significant when it was below 0.002.

General Linear Model - Analysis of Covariance	AD 10^−3^ mm^2^/s		RD 10^−3^ mm^2^/s		ADC 10^−3^ mm^2^/s		FA	
	Left	Right	Left	Right	Left	Right	Left	Right
Girls vs. Boys								
Thalamo-cortical	0.297	0.255	0.270	0.245	0.271	0.231	0.497	0.536
Thalamus	0.950	0.803	0.851	0.597	0.881	0.832	0.789	0.130
Putamen	0.351	0.401	0.639	0.990	0.440	0.702	0.584	0.376
Caudate Nucleus	0.720	0.440	0.415	0.632	0.481	0.506	0.615	0.695
Globus-Pallidus	0.207	0.324	0.269	0.166	0.191	0.174	0.763	0.556

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
