# Peer review of "Sexual Dimorphisms and Asymmetries of the Thalamo-Cortical Pathways and Subcortical Grey Matter of Term Born Healthy Neonates: An Investigation with Diffusion Tensor MRI"

_diagnostics, 2021, doi:10.3390/diagnostics11030560_

Round 1
Reviewer 1 Report
The paper presents interesting investigations of the thalomo-cortical pathwey in term born neonates with using Diffusion tensor MRI. The authors propose normative data estimating the proper development of the white matter integrity in brain development. However, disadvantage of this work is too small and poorly defined research group of term born neonates; one may have doubts about statistical analyzes due to boys (15) and girls(13). In my opinion, conclusion that “structural impairment of the thalamic asymmetry might be a strong objective marker of cognitive and motor functioning in children” should be based on studies conducted on a larger population of neonates and confirmed or compared with other test methods. Although I realize that research is not easy to perform. In addition, the paper seems to be very detailed but is not very readable.
Author Response
Reviewer #1
The paper presents interesting investigations of the thalomo-cortical pathway in term born neonates with using Diffusion tensor MRI. The authors propose normative data estimating the proper development of the white matter integrity in brain development.
- However, disadvantage of this work is too small and poorly defined research group of term born neonates; one may have doubts about statistical analyzes due to boys (15) and girls(13).
We agree with this observation. While the strength of this study is the confine of high technology to healthy born neonates, the limitation is the sample size. We could extend the duration of the study to recruit more subjects, but the trade-off is an upgrade of the scanner that might bias the measured data. We preferred to move forward with this sample size instead.
- In my opinion, conclusion that “structural impairment of the thalamic asymmetry might be a strong objective marker of cognitive and motor functioning in children” should be based on studies conducted on a larger population of neonates and confirmed or compared with other test methods. Although I realize that research is not easy to perform.
We thank the reviewer for this remark, and we do share the same thoughts. The ideal research would be a longitudinal study where we combine this MRI protocol with neurobehavioral exams in order to identify the biological link with the neurodevelopment outcome. For this reason, we rephrased our suggestion as follow.
Although this is study has limited number of subjects, we suggest that structural impairment of the thalamic asymmetry might be an objective marker of cognitive and motor functioning children. Further investigations involving both larger sample size and multimodal imaging are required to confirm this suggestion.
- In addition, the paper seems to be very detailed but is not very readable.
We revisited the main document and in particular the Discussion section to make easier to red. Our modifications were added in red color.

Reviewer 2 Report
This is a relativly interesting study showing some insights into developing brain. It might be of potential interest to readers, but some issues should be adressed.
What was the medical justification for MRI on healthy infant ?
Were there any complications or advers events of the precedure ?
If the ROI drawings were done by 1 or more researcher ? Was intraobserver coefficiant calculated ?
Table 1 should be extensively clarified. Arrebiations such as "W" or "D" are not clear nor pleasent to the eye. Most readears go through figures and tables alone. They should be self-explanatory
Font size on Figures 2-5 should be enlarged. Axis Y and X should be clearly named.
In Discussion some issues should be adressed:
- Wheter it is possible that the rusluts would change after myelinization is complete ?
- Would authors expect ther results to be different after milstones in neurological occure ?
- What is the reason of lack of asymtery in globus pallidus ?
Author Response
Reviewer II
This is a relatively interesting study showing some insights into developing brain. It might be of potential interest to readers, but some issues should be addressed.
- What was the medical justification for MRI on healthy infant?
The healthy term born neonates presented here were part of a prospective clinical trial investigating neonates with congenital heart disease undergoing transposition of the great artery.
- Were there any complications or adverse events of the procedure?
We appreciate the thought raise here by the reviewer. We did not report any adverse events or complication and we did not administrate any medicine or sedative. As mentioned in the paragraph “2.1 Subjects “these neonates were monitored during the MRI exam, and a research nurse was dedicated to check their oxygen saturation, the arterial blood pressure, the heart rate, and PCO2. In addition, their mother was allowed to stay around to observe and intervene whenever there is a need for breast-feeding.
- If the ROI drawings were done by 1 or more researcher? Was intra-observer coefficient calculated?
This is an excellent remark and we are thankful for asking about the intra-observer. We neglected to include this in the section “2.2.1 Image processing”. We added the following sentences:
Experienced scientist with over 10 years of practice carried out the ROI drawings. This procedure was repeated twice (one month apart) to minimize the subjectivity error. The results of the two measures were averaged to perform the statistical comparisons (Below a screenshot of the measurements performed on each structure).
- Table 1 should be extensively clarified. Abbreviations such as "W" or "D" are not clear nor pleasant to the eye. Most readers go through figures and tables alone. They should be self-explanatory
Many thanks for this reminder and our apology for the missing details. We modified the legend so one can read.
Table 1: Descriptive gestation age (GA) statistics of the neonates per group (W refers to number of week and D number of days).
- Font size on Figures 2-5 should be enlarged. Axis Y and X should be clearly named.
We agree with this remark. All figures from 2 to 5 were revisited and we labeled the Y axis by the category and the X axis by the variable. All abbreviations were changed and named (axial diffusion, radial diffusion or apparent diffusion instead of AD, RD, and ADC). Please see an example given below:
- In Discussion some issues should be addressed:
- Whether it is possible that the results would change after myelination is complete ? Would authors expect their results to be different after millstones in neurological occur ?
The reviewer is right, and we appreciate this question as we did not emphasize on this point. We included the following sentence in the discussion to highlight the changes of the DTI indices values with age.
In full-term neonates GA, birth weight, crown-heel length and head circumference affect WM maturation and it has been demonstrated that there is biological link between the DTI indices (FA, ADC, AD, and RD) and brain WM microstructural integrity, water diffusivities, axonal growth and myelination [Rose (2015), Dubois (2014)]. The thalamus and the basal-ganglia experience significant changes with age and they all reach the 90% mark between the ages of 21 and 24 years (Lebel C, 2008).
The reported structural asymmetries are in line with previous findings showing that the correlation of WM with GA is more pronounced in the left hemisphere Jin et al. (2019) and the lateralization of the somatosensory can be observed at birth (Erberich et al., 2006). The lateralization might be attributed to genetic programs during prenatal stage, as asymmetry of gene expression in the human embryonic cortex has been found at gestational of 12 weeks (Erberich et al., 2006).
- What is the reason of lack of asymmetry in globus pallidus?
The following statement was included in the “Discussion” section that may help understanding the findings.
During the neonate period, such changes in DTI metrics correlate with any of the following phenomena: increase in myelin sheaths, increments of axon density, cell membrane, and glial cells (Dubois et al., 2014). The left and right globus-pallidus reflects development of postural balance However, (Cahill-Rowley et al,) did not report a significant correlation between the DTI metrics of the globus pallidus and both walking velocity and single limb support. The lack of significant asymmetry observed in the globus-pallidus may presumably relate to slow development of sensorimotor ability. This feature might change with age and neurodevelopmental process but given the complexity in brain development and the environmental relate, further investigation is needed to explain the physiological and neurological processes as deep gray matter structures show very large percent changes and the white matter structures showing smaller, but still significant, changes (Lebel C, 2008)
Additional References
45) Lebel C, Walker L, et al. Microstructural maturation of the human brain from childhood to adulthood NeuroImage 40 (2008) 1044–1055
46) Dubois J., Dehaene-Lambertz G., et al. The early development of brain white matter: A review of imaging studies in fetuses, newborns and infants. Neuroscience (2014), 276, 48–71.
47) Rose J, Cahill-Rowley K, et al. Neonatal brain microstructure correlates of neurodevelopment and gait in preterm children 18–22 mo of age: an MRI and DTI study. Pediatric Researc h Volume 78 | Number 6 | December 2015
73) Cahill-Rowley K, Schadl K, et al. Prediction of Gait Impairment in Toddlers Born Preterm From Near-Term Brain Microstructure Assessed With DTI, Using Exhaustive Feature Selection and Cross-Validation Frontiers in Human Neuroscience, Sept 2019, Volume 13, Article 305
75) Jin C, Li Y, Li X, et al. Associations of gestational age and birth anthropometric indicators with brain white matter maturation in full-term neonates. Hum Brain Mapp. 2019;40:3620–3630.
76) Erberich, S., Panigrahy, A., et al. Somatosensory lateralization in the newborn brain. NeuroImage (2006), 29(1), 155–161.

Round 2
Reviewer 1 Report
After the corrections have been made , the artilce can be published despite the fact that the obtained results were not compared with others (probably due to the lack of literature data).